# MedREK: Retrieval-Based Editing for Medical LLMs with Key-Aware Prompts

## Abstract

LLMs hold great promise for healthcare applications, but the rapid evolution of medical knowledge and errors in training data often cause them to generate outdated or inaccurate information, limiting their applicability in high-stakes clinical practice. Model editing has emerged as a potential remedy without full retraining. While parameter-based editing often compromises locality and is thus ill-suited for the medical domain, retrieval-based editing offers a more viable alternative. However, it still faces two critical challenges: (1) representation overlap within the medical knowledge space often causes inaccurate retrieval and reduces editing accuracy; (2) existing methods are restricted to single-sample edits, while batch-editing remains largely unexplored despite its importance for real-world medical applications. To address these challenges, we first construct MedVersa, an enhanced benchmark with broader coverage of medical subjects, designed to evaluate both single and batch edits under strict locality constraints. We then propose MedREK, a retrieval-based editing framework that integrates a shared query-key module for precise matching with an attention-based prompt encoder for informative guidance. Experimental results on various medical benchmarks demonstrate that our MedREK achieves superior performance across different core metrics and provides the first validated solution for batch-editing in medical LLMs.

## 1 Introduction

The remarkable success of large language models (LLMs) (Touvron et al., 2023a;b; Bai et al., 2023) in recent years has attracted significant attention from the medical community, leading to the emergence of specialized medical LLMs such as BioGPT (Luo et al., 2022), Med-PaLM (Singhal et al., 2023), ChatDoctor (Li et al., 2023) and PMC-LLaMA (Wu et al., 2024). However, due to the rapid evolution of medical knowledge and limitations in training data, these models may generate inaccurate or even fabricated responses (hallucinations), which can be particularly harmful in real-world medical advising and decision-making scenarios (Tian et al., 2024; He et al., 2025).

To address this issue, model editing (Meng et al., 2022; 2023; Wang et al., 2024) has emerged as a promising approach for updating the knowledge of pre-trained LLMs without requiring full retraining. Despite its growing popularity, model editing remains underexplored in the medical domain. A pioneering effort, MedLaSA (Xu et al., 2024), follows the locate-then-edit paradigm and constructs medical benchmarks for evaluating single-edit scenarios. However, its benchmark remains confined to simple single-edit settings and overlooks the more realistic batch-edit scenario. Moreover, locate-then-edit methods, which modify a small subset of parameters, often induce side effects on unrelated knowledge and compromise locality (Fang et al., 2025). This limitation is particularly concerning in medical applications, where reliability and consistency are critical.

To enable more realistic evaluation, we introduce MedVersa, the first benchmark designed to explore batch-editing in medical scenarios. It better reflects real-world use cases in which multiple related facts require simultaneous updates. We also substantially broaden subject coverage, yielding a more comprehensive benchmark that facilitates future research on knowledge editing for medical LLMs. Under batch-editing scenarios, we investigate retrieval-based approaches (Huang et al., 2023a; Yu et al., 2024) as an alternative to parameter-based editing, since they store new knowledge in an external memory module without altering the model's original parameters and thereby better preserve locality. However, our preliminary experiments show that the state-of-the-art retrieval-based method

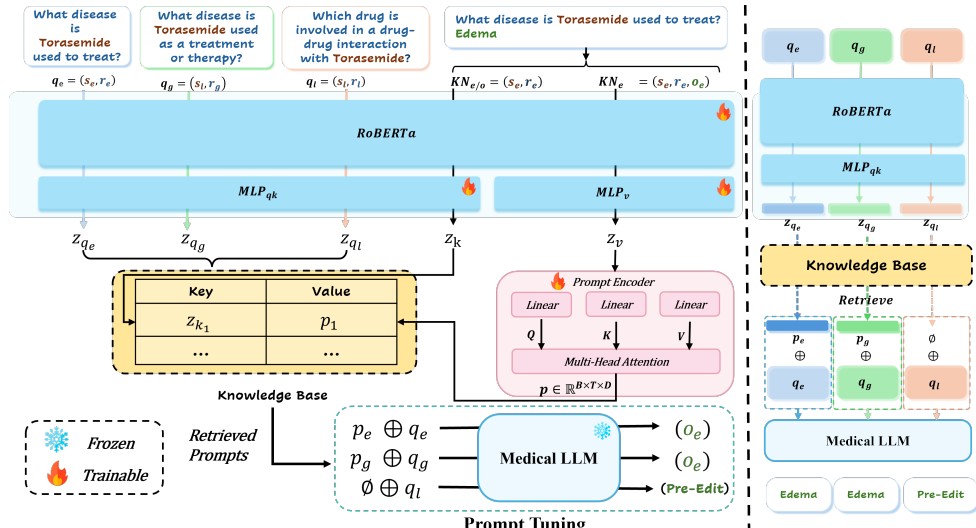

Figure 1: Pipeline of our MedREK. Left: Construction of the knowledge base by encoding medical knowledge into key-value pairs. Right: Inference process where different types of queries are encoded to retrieve relevant knowledge and generate attention-based prompts for precise model editing.

RECIPE (Chen et al., 2024b) often fails to retrieve the correct factual entries on the challenging medical evaluations. Further analysis indicates that the medical domain contains many textually similar but factually distinct knowledge items, which causes *representation overlap* in the retrieval space and thus hinders accurate knowledge matching. Based on these observations, we propose **Med**ical **R**etrieval-based **E**diting with **K**ey-aware prompts (**MedREK**).

As shown in Fig. 1, our method introduces two key components: (1) *a shared query-key MLP*, which unifies the representation space of queries and keys for more precise knowledge retrieval; (2) *an attention-based prompt encoder*, which generates more informative prompts to guide editing. Building on these components, our MedREK achieves strong performance in both single-edit and batch-edit evaluations. Comprehensive experiments on various medical benchmarks further demonstrate that our MedREK achieves state-of-the-art performance across *Efficacy*, *Generality*, and *Locality* metrics, validating the effectiveness of the proposed *shared query-key MLP* and *attention-based prompt encoder*. In summary, our main contributions are as follows:

- We construct MedVersa, a medical factual knowledge editing benchmark that enables a more realistic batch-editing setting and offers broader subject coverage than existing benchmarks.
- We are the first to explore retrieval-based model editing for medical LLMs, proposing two novel components that separately enhance key-query alignment and prompt quality.
- We propose MedREK, a novel method that markedly improves knowledge editing performance by enhancing key-query alignment. Extensive experiments confirm that MedREK achieves state-of-the-art results across core metrics, demonstrating particularly strong gains in locality.

## 2 RELATED WORKS

**Model Editing** aims to efficiently update a pre-trained model's behavior in response to new or corrected knowledge, without full retraining or negatively affecting unrelated predictions. Existing methods fall into three categories: locate-then-edit methods (Meng et al., 2022; Li et al., 2024), meta-learning-based strategies (Tan et al., 2024; Mitchell et al., 2022a), and retrieval-based approaches (Wang et al., 2024; Chen et al., 2024b).

- *Locate-then-edit methods*, such as ROME (Meng et al., 2022) and MEMIT (Meng et al., 2023), identify the parameters associated with specific knowledge in LLMs and directly modify them to incorporate new information.
- *Meta-learning-based methods* take a different route by manipulating gradients to perform global parameter updates, aiming for more generalizable knowledge integration. For example, both

KE (Cao et al., 2021) and MEND (Mitchell et al., 2022a) employ a lightweight hyper-network to adjust the updating gradient. InstructEdit (Wang et al., 2023) builds on MEND by introducing instruction tuning for handling various tasks. Moreover, MALMEN (Tan et al., 2024) further generalizes MEND to batch-editing under memory constraints.

- *Retrieval-based approaches* edit model output by storing new knowledge in external memory modules, without altering pre-trained weights. These modules can take the form of codebooks, neurons, or auxiliary models, as demonstrated in SERAC (Mitchell et al., 2022b), T-Patcher (Huang et al., 2023a), GRACE (Hartvigsen et al., 2023), and MELO (Yu et al., 2024). More recently, WISE (Wang et al., 2024) designs dual memory and routes the model to the side memory in FFN for editing. RECIPE (Chen et al., 2024b) utilizes continuous prompt learning to prefix knowledge to the input query and dynamically retrieves knowledge from the knowledge base.

Accurate and up-to-date medical knowledge is critical for the safe and reliable application of large language models in healthcare. As medical facts evolve rapidly with new research and clinical guidelines, the ability to edit existing knowledge without retraining is essential. Given the importance of medical knowledge editing, MedLaSA (Xu et al., 2024) pioneers this area and introduces the MedCF benchmark, but direct model edits can degrade locality and are limited to single-edit protocols. Building on this medical model editing line, we extend the setting to batch-editing and adopt the retrieval-based strategy. We further identify medical domain retrieval failures and remedy them with targeted improvements, culminating in MedREK, which delivers more reliable knowledge updates.

**Prompt Tuning.** As a representative parameter-efficient fine-tuning technique, prompt tuning is widely used in adapting foundation models to downstream tasks. It includes discrete prompts, expressed as actual text strings, and continuous prompts, encoded directly within the embedding space of the language model. Specifically, discrete prompts (Shin et al., 2020; Schick & Schütze, 2021a;b; Gao et al., 2021) are manually designed or automatically searched to elicit desired behavior from the model, while continuous prompts (Li & Liang, 2021; Liu et al., 2022; 2023; Xu et al., 2023) are trainable embeddings learned through optimization, capable of capturing more nuanced task-specific information. In this work, we train a prompt encoder to generate a continuous prompt that serves as external memory and enables targeted medical knowledge editing. These lightweight prompts can be stored efficiently, enabling effective editing with minimal resource overhead.

# 3 MOTIVATION

## 3.1 PRELIMINARIES

**Model Editing.** Let $f_\theta \in \mathcal{F} : \mathcal{Q} \mapsto \mathcal{A}$ denote the large language model which can map an input query $q \in \mathcal{Q}$ to the output answer $a = f_\theta(q)$. Given an edit sample pair $(q_e, o_e)$ that $f_\theta(q_e) \neq o_e$, model editing hopes to modify $f_\theta$ to $f'_\theta$ so that,

$$f'_\theta = \text{ME}(f_\theta, q_e, o_e), \quad f'_\theta(q_e) = o_e, \tag{1}$$

where $\text{ME}(\cdot, \cdot, \cdot)$ means the model editor.

For a factual knowledge triple $\text{KN}_e = (s_e, r_e, o_e)$, the components $s_e$, $r_e$, and $o_e$ denote the subject, relation, and object, respectively (Mallen et al., 2023). We define the pair $\text{KN}_{e/o} := (s_e, r_e)$ as the pre-defined knowledge key, and thus $\text{KN}_e = (\text{KN}_{e/o}, o_e)$.

**Evaluation Metrics.** Beyond accurate knowledge editing (*i.e.*, $f'_\theta(q_e) = o_e$), an ideal model editor should meet a range of additional requirements, each evaluated through specific metrics (Huang et al., 2023b; Yao et al., 2023; Zhang et al., 2024).

*Generality.* The edited model $f'_\theta$ is expected to generalize beyond the editing query $q_e$ to correctly answer similar questions $q_g$.

$$\mathbb{E}_{(q_g, o_e)} \, \mathbb{I}\{f'_\theta(q_g) = o_e\}, \tag{2}$$

*Locality.* The edited model $f'_\theta$ should preserve the overall capabilities of the original large language model (*i.e.*, $f_\theta$) by ensuring that examples unrelated to the editing target remain unaffected.

$$\mathbb{E}_{(q_l, o_l)} \, \mathbb{I}\{f'_\theta(q_l) = f_\theta(q_l) = o_l\}, \tag{3}$$

where $(q_l, o_l)$ denotes question-answer pairs that are semantically irrelevant to the edited samples $(q_e, o_e)$ and should ideally remain unaffected. Note that, given the high precision requirements in medical domains, this metric becomes particularly critical when performing knowledge editing.

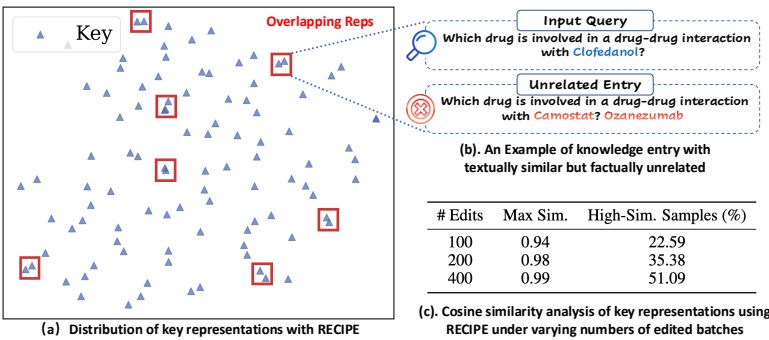

Figure 2: (a) Visualization of key representations in retrieval-based editing methods. Key overlap raises retrieval errors, decreasing editing accuracy. (b) Example of incorrect retrieval caused by key overlap. (c) Cosine similarity measurements quantifying degree of overlap among key representations.

*Efficacy.* This metric calculates the accuracy of the modified model $f'_\theta$ on the edited examples $(q_e, o_e)$, thereby reflecting the effectiveness of the edit.

$$\mathbb{E}_{(q_e, o_e)} \, \mathbb{I}\{f'_\theta(q_e) = o_e\}. \tag{4}$$

*Fluency.* This metric is designed to reflect the linguistic coherence of the response. It is quantified by computing the weighted average of bi-gram and tri-gram entropies, given by:

$$\sum_k f_n(k) \, \log_2 f_n(k), \tag{5}$$

where $f_n(\cdot)$ is the $n$-gram frequency distribution, and $k$ denotes a model-generated output.

**Editing Setting.** Single-editing refers to modifying one knowledge item at a time, while batch-editing involves applying multiple edits simultaneously. Real-world updates such as revised treatment guidelines, newly discovered drug interactions, or retracted clinical findings often affect multiple related facts simultaneously. This makes batch-editing a realistic and practically valuable setting. However, within medical domains, current approaches are primarily designed for single-edit scenarios, leaving the more realistic challenge of batch-editing largely unexplored. This further highlights the necessity of more precise retrieval mechanisms.

### 3.2 LOCATE-THEN-EDIT METHODS HARM LOCALITY

Locate-then-edit approaches (Meng et al., 2023; 2022) typically use causal tracing to identify influential model components and modify the corresponding parameters. However, such parameter updates often introduce undesirable side effects on unrelated knowledge, leading to a notable drop in locality metrics (Fang et al., 2025). This makes them ill-suited for medical applications, where high locality (i.e., preserving irrelevant knowledge) is essential to ensure reliability and safety. To mitigate these issues, we turn our attention to the alternative retrieval-based methods, which avoid direct parameter modification and generally offer better locality preservation.

### 3.3 RETRIEVAL-BASED METHODS STRUGGLE WITH OVERLAPPING KNOWLEDGE

In medical scenarios, many knowledge entries may be *textually similar but factually unrelated*, making accurate retrieval especially challenging. This calls for a more precise matching mechanism between queries and stored knowledge. In particular, competing retrieval-based methods, such as RECIPE (Chen et al., 2024b), fail to retrieve the correct factual entry from the knowledge base on the medical knowledge editing benchmark. As shown in Figure 2 (a), the representation space of RECIPE exhibits significant overlap among distinct knowledge items, leading to frequent confusion. This issue is exemplified in Figure 2 (b): for the query "Which drug is involved in a drug-drug interaction with Clofedanol?", the model incorrectly retrieves the unrelated entry "Which drug is involved in a drug-drug interaction with Camostat? Ozanezumab" from the knowledge base, due to overlapping key representations.

To quantify this phenomenon, Figure 2 (c) further measures the overlap among key representations. Specifically, we report the percentage of unique samples involved in at least one pair with cosine

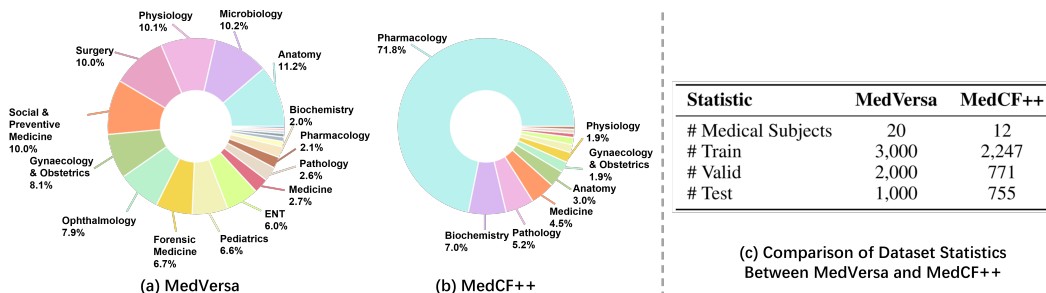

Figure 3: Comparison of MedVersa and MedCF++ in terms of medical subjects and dataset statistics. We provide the full distribution of medical subjects in the two datasets in Appendix.

similarity $> 0.6$, denoted as "High-Sim. Samples (%)", which reflects how many representations are highly aligned with others in the same batch. It can be observed that, as batch size increases, the share of samples participating in at least one high-similarity pair rises, reaching 51.09%, indicating greater representation overlap in larger batches. This provides quantitative support for our motivation and underscores the need for more precise retrieval tailored to medical knowledge editing.

## 4 METHOD

To address the limitations discussed above, we first construct MedVersa (*i.e.*, **Med**ical **Versa**tile Knowledge Editing Dataset), an enhanced benchmark with broader coverage of medical subjects, designed to systematically examine batch-edit scenarios in the medical domain. We then present the **MedREK** algorithm (*i.e.*, **Med**ical **R**etrieval-based **E**diting with **K**ey-aware prompts), designed to mitigate knowledge forgetting from direct parameter updates and effectively tackle the issue of irrelevant retrievals highlighted in Figure 2.

### 4.1 CONSTRUCTION OF MEDVERSA DATASET

The current MedCF (Xu et al., 2024) benchmark–built on the Drug Repurposing Knowledge Graph (DRKG) that links compounds, diseases, biological processes, side effects, and symptoms– presents two limitations: (i) Some prompts are reused for both Efficacy and Locality across the train/validation/test

Table 1: An example from the MedVersa dataset.

| Medical Versatile Knowledge Editing Dataset |
| --- |
| **Efficacy Question**: What is the treatment for multiple carboxylase deficiency? |
| **Generality Question**: What is the therapeutic management for multiple carboxylase deficiency? |
| **Locality Question**: At what age does purposeful movement typically start in infants? |
| **Ground Truth**: Biotin |
| **Counterfactual Edit Target**: Thiamine |
| **Locality Ground Truth**: 6 months |
| **Efficacy QA Pair**: (Efficacy Question, Counterfactual Edit Target) |
| **Generality QA Pair**: (Generality Question, Counterfactual Edit Target) |
| **Locality QA Pair**: (Locality Question, Locality Ground Truth) |

splits, which confounds batch-edit evaluation because edits targeting the Efficacy answer should not influence unrelated Locality responses. (ii) MedCF (Xu et al., 2024) remains limited in its topical breadth: most selected entities come from drugs and compounds, concentrating the benchmark in pharmacology. To address these issues, we construct MedVersa built on MedMCQA (Pal et al., 2022), which eliminates Efficacy–Locality prompt overlap to enable reliable batch-edit assessment and broadens coverage across medical subjects for more comprehensive evaluation. For clarity, the intermediate version that fixes only limitation (i) is denoted as MedCF++. An illustrative example is shown in Table 1, and additional construction details are provided in Appendix B and C.

**Comparison of MedVersa and MedCF++.** In Figure 3, we present a detailed comparison of MedVersa and MedCF++. MedVersa spans a broader range of 20 medical subjects, including areas absent in MedCF++ such as Pediatrics and Social & Preventive Medicine. In contrast, MedCF++ covers only 12 medical subjects, with Pharmacology accounting for the majority (71.8%). The broader and more balanced subject coverage of MedVersa fills the gap in evaluating medical knowledge editing across a wider range of medical domains, enabling more comprehensive assessment.

### 4.2 THE PROPOSED MEDREK ALGORITHM

As shown in Figure 1, our proposed MedREK comprises two main components: ❶ a representation model featuring a unique *shared query-key MLP* (*i.e.*, $\mathrm{MLP}_{qk}$) designed for effective knowledge retrieval, and ❷ *an attention-based prompt encoder* for generating more informative prompts.

### 4.2.1 The Shared Query-Key MLP

To ensure precise retrieval in model editing, we design a retrieval mechanism that satisfies both relevance and selectivity. Specifically, it aims to (1) accurately retrieve the specific knowledge associated with the given query, which reflects the Efficacy and Generality criteria, and (2) avoid unnecessary retrieval when the query is unrelated to stored knowledge, which corresponds to the locality requirement. To this end, we employ *a shared query-key MLP* encoder that encodes both memory keys and incoming queries into a unified representation space. Importantly, by constructing keys solely from the subject–relation pair $(s_i, r_i)$ instead of the full triplet $(s_i, r_i, o_i)$, our design avoids incorporating irrelevant object information, thereby reducing retrieval noise.

Given an input sentence (either in the form of queries $q_e$, $q_g$ or $q_l$, knowledge key $\mathrm{KN}_{e/o}$ or a full knowledge triplet $\mathrm{KN}_e$), we first tokenize it using a pre-trained RoBERTa (Liu et al., 2019) tokenizer and obtain contextualized embeddings from the last hidden layer:

$$\mathbf{H} \in \mathbb{R}^{L \times d}, \quad \mathbf{p} \in \mathbb{R}^d \tag{6}$$

where $L$ is the sequence length and $d$ is the hidden dimension. We aggregate these embeddings by concatenating the CLS token embedding $\mathbf{p}$ with pooled statistics (mean, max, and min):

$$\mathbf{x} = [\mathbf{p}; \mathrm{mean}(\mathbf{H}); \mathrm{max}(\mathbf{H}); \mathrm{min}(\mathbf{H})] \in \mathbb{R}^{4d}. \tag{7}$$

Next, this vector $\mathbf{x}$ is fed into the representation model's MLP layers. For queries $q_e$, $q_g$ or $q_l$ and knowledge key $\mathrm{KN}_{e/o}$, we apply a shared MLP encoder $\mathrm{MLP}_{\mathrm{qk}}$:

$$\mathbf{z}_q = \mathrm{MLP}_{\mathrm{qk}}(\mathbf{x}) = \mathrm{ReLU}(\mathbf{W}_{q2}(\mathbf{W}_{q1}\mathbf{x})) + \mathbf{W}_{q1}\mathbf{x}, \tag{8}$$

$$\mathbf{z}_k = \mathrm{MLP}_{\mathrm{qk}}(\mathbf{x}) = \mathrm{ReLU}(\mathbf{W}_{q2}(\mathbf{W}_{q1}\mathbf{x})) + \mathbf{W}_{q1}\mathbf{x}, \tag{9}$$

which projects them into a shared representation space for accurate matching in the retrieval stage. For full knowledge triplets $\mathrm{KN}_e$ used in prompt generation, a separate MLP encoder $\mathrm{MLP}_v$ is applied:

$$\mathbf{z}_v = \mathrm{MLP}_v(\mathbf{x}) = \mathrm{ReLU}(\mathbf{W}_{k2}(\mathbf{W}_{k1}\mathbf{x})) + \mathbf{W}_{k1}\mathbf{x}, \tag{10}$$

where $\mathbf{z}_k$ is stored as the key of the $(k, v)$ pair in the knowledge base, while $\mathbf{z}_v$ is further passed to the prompt encoder to generate continuous prompt as discussed next.

### 4.2.2 Attention-Based Prompt Encoder

To enable precise edits, we further design a prompt encoder that maps each knowledge representation into a sequence of continuous prompt tokens via multi-head attention. Unlike traditional prompt tuning with fixed prompts for broad tasks, medical knowledge editing demands fine-grained, fact-specific modifications. Given the subtle differences between medical facts, dynamically generating prompts conditioned on the input knowledge is essential. Our prompt encoder learns to produce knowledge-specific prompts, leading to more accurate and effective editing. Specifically, given the knowledge representation $\mathbf{z}_v \in \mathbb{R}^{d_{in}}$ obtained from the representation model, we first project it into a set of query vectors for prompt tokens, and a single key and value vector:

$$\mathbf{Q} = \mathrm{reshape}(\mathbf{W}_q \mathbf{z}_v) \in \mathbb{R}^{T \times d}, \quad \mathbf{K} = \mathbf{W}_k \mathbf{z}_v \in \mathbb{R}^{d \times 1}, \quad \mathbf{V} = \mathbf{W}_v \mathbf{z}_v \in \mathbb{R}^{d \times 1},$$

where $T$ is the number of prompt tokens, $d_{\mathrm{in}}$ and $d$ denote input and output dimensions, and $\mathbf{W}q \in \mathbb{R}^{Td \times d_{\mathrm{in}}}$, $\mathbf{W}_k, \mathbf{W}v \in \mathbb{R}^{d \times d_{\mathrm{in}}}$ are learnable parameters in the attention mechanism, following the standard formulation in (Vaswani et al., 2017). Next, a multi-head attention module is used to allow each prompt token to attend to the same knowledge vector and obtain contextualized prompt representations:

$$\mathbf{p} = \mathrm{MultiHeadAttn}(\mathbf{Q}, \mathbf{K}, \mathbf{V}) \in \mathbb{R}^{T \times d}. \tag{11}$$

In this way, $\mathbf{p}_{\mathrm{final}} \in \mathbb{R}^{B \times T \times d}$, which is stored as the value of the $(k, v)$ pair in the knowledge base. The number of continuous prompt tokens $T$ is a hyper-parameter, and we describe its selection for different datasets in Section 5.1.

### 4.3 Retrieval Pipeline and Training

**Retrieval Pipeline.** We employ a trainable knowledge prototype representation $\mathbf{z}_{pt}$ as a dynamic threshold for retrieval in the representation model. During inference, retrieval is performed before the query tokens are fed into the embedding layer:

Table 2: The overall results for single-editing (# Editing=1) and batch-editing (# Editing>1) using Meditron-7B on MedCF++ and MedVersa datasets.

| # Editing | Method | MedCF++ | | | | | | | | MedVersa | | | | |
|---|---|---|---|---|---|---|---|---|---|---|---|---|---|---|
| | | Eff. | Gen. | Loc. | | | | Flu. | Avg. | Eff. | Gen. | Loc. | Flu. | Avg. |
| | | | | TD | EM | SS | TS | | | | | | | |
| 1 | MEND | 26.60 | 28.14 | 89.03 | 90.47 | 88.73 | 87.13 | 575.98 | 58.10 | 27.09 | 28.65 | 94.24 | 583.09 | 61.05 |
| | MEMIT | 79.06 | 69.23 | 98.02 | 77.40 | 97.65 | 93.57 | 562.91 | 82.90 | 70.02 | 46.81 | 99.42 | 575.86 | 78.92 |
| | MedLaSA | 72.16 | 68.84 | 80.11 | 85.13 | 80.03 | 79.54 | 578.48 | 75.85 | 74.30 | 71.57 | 87.01 | 555.69 | 79.97 |
| | RECIPE | 72.66 | 77.12 | 92.64 | 99.80 | 90.29 | 90.59 | 586.68 | 84.11 | 57.89 | 57.44 | 99.03 | 599.66 | 78.35 |
| | Ours | 78.50 | 80.61 | 99.42 | 98.96 | 99.34 | 98.67 | 587.00 | 89.33 | 74.49 | 70.46 | 100.00 | 579.63 | 86.24 |
| 10 | MEND | 27.48 | 28.18 | 76.24 | 79.57 | 77.55 | 76.73 | 579.21 | 52.68 | 28.52 | 29.85 | 86.30 | 585.75 | 57.74 |
| | MEMIT | 79.24 | 69.81 | 94.93 | 75.99 | 93.88 | 89.34 | 562.82 | 81.53 | 66.60 | 46.25 | 97.41 | 574.47 | 76.92 |
| | RECIPE | 72.09 | 76.46 | 89.06 | 96.07 | 88.17 | 87.81 | 586.29 | 82.28 | 57.89 | 57.41 | 96.72 | 600.27 | 77.18 |
| | Ours | 78.52 | 80.63 | 99.33 | 98.89 | 99.35 | 98.51 | 589.56 | 89.30 | 74.49 | 70.46 | 99.90 | 600.17 | 86.19 |
| 50 | MEND | 25.05 | 25.80 | 62.99 | 61.33 | 64.78 | 60.25 | 578.74 | 43.88 | 27.29 | 28.07 | 68.97 | 583.66 | 48.33 |
| | MEMIT | 75.59 | 67.08 | 90.14 | 77.04 | 89.53 | 85.99 | 562.07 | 78.51 | 68.98 | 48.51 | 93.04 | 573.18 | 75.89 |
| | RECIPE | 70.33 | 73.82 | 78.75 | 87.20 | 79.15 | 76.48 | 589.75 | 76.23 | 57.89 | 57.44 | 89.65 | 599.59 | 73.66 |
| | Ours | 78.54 | 80.61 | 98.55 | 98.70 | 98.80 | 97.87 | 589.79 | 89.03 | 74.49 | 70.39 | 99.68 | 599.25 | 86.06 |
| 100 | MEND | 24.88 | 25.14 | 65.03 | 62.49 | 66.24 | 66.24 | 578.46 | 44.52 | 26.07 | 26.26 | 61.43 | 577.92 | 43.80 |
| | MEMIT | 76.36 | 66.42 | 87.91 | 76.74 | 87.26 | 83.52 | 562.53 | 77.62 | 70.26 | 49.20 | 89.73 | 573.27 | 74.73 |
| | RECIPE | 68.30 | 70.74 | 72.74 | 79.73 | 72.48 | 71.58 | 588.21 | 71.83 | 57.89 | 57.37 | 84.19 | 600.31 | 70.91 |
| | Ours | 77.96 | 79.88 | 97.76 | 98.20 | 97.57 | 97.19 | 588.01 | 88.30 | 74.49 | 70.46 | 99.45 | 598.67 | 85.96 |

$$f_r(q) = \begin{cases} \mathbf{p}_i, & \text{if } \mathbf{z}_q \cdot \mathbf{z}_{k_i} > \mathbf{z}_q \cdot \mathbf{z}_{pt} \\ \varnothing, & \text{otherwise} \end{cases} \quad (12)$$

where $f_r(\cdot)$ denotes the retrieval process for the query $q$, and $\mathbf{z}_{k_i}$ is the most similar key representation in the knowledge base. A prompt is retrieved only if it is more similar to the query than the learned prototype. If no prompt is returned, the model proceeds as usual, with inference unaffected.

**Training.** Following RECIPE (Chen et al., 2024b), we utilize the training loss as $\mathcal{L}_{\text{total}} = \mathcal{L}_{\text{contra}} + \mathcal{L}_{\text{edit}}$, where $\mathcal{L}_{\text{edit}} = \frac{1}{B} \sum_{i=1}^{B} \left( \mathcal{L}_{\text{eff}}^{(i)} + \mathcal{L}_{\text{gen}}^{(i)} + \mathcal{L}_{\text{loc}}^{(i)} \right)$. Details of each sub-loss are provided in Appendix E.

## 5 EXPERIMENTS

In this section, we conduct experiments to answer the following research questions: **RQ1:** Does MedREK outperform strong baseline editors on medical LLMs across core metrics and under batch-editing? **RQ2:** Do the proposed modules contribute significant gains individually and jointly? **RQ3:** Is the retrieval mechanism effective at locating the correct knowledge?

### 5.1 EXPERIMENTAL SETUP

**Settings & Benchmarks.** We evaluate model editors under both *Single-editing* and *Batch-editing* settings to comprehensively assess their robustness and generalization capabilities. Experiments are conducted on the improved MedCF++ benchmark and our newly constructed MedVersa dataset. Unlike prior work (Xu et al., 2024), which only considers single-editing, we explore more realistic large-scale update scenarios by evaluating performance under 10/50/100-edit configurations.

**Implementation Details.** Following MedLaSA (Xu et al., 2024), we use LLaMA2-based (Touvron et al., 2023c) Meditron-7B (Li et al., 2023) as the primary model and include LLaMA3-based (Grattafiori et al., 2024) HuatuoGPT-o1-8B (Chen et al., 2024a) for additional evaluation. We train MedREK for 200 epochs and report the results using the checkpoint with the smallest loss. We use 3 and 8 prompt tokens for MedCF++ and MedVersa, respectively. See Appendix A for details of hyper-parameters.

**Baselines.** We compare MedREK with several knowledge editing baselines, including MEND (Mitchell et al., 2022a), MEMIT (Meng et al., 2023), MedLaSA (Xu et al., 2024), and RECIPE (Chen et al., 2024b). Note that MedLaSA (Xu et al., 2024) is excluded from batch-editing experiments, as its parameter modification strategy is inherently designed for single-edit settings and cannot be directly extended to handle multiple simultaneous edits. This limitation underscores the importance of developing batch-capable editing methods. We re-implement RECIPE in the medical domain for fair comparison.

**Evaluation metrics.** In line with MedLaSA (Xu et al., 2024), we adopt four evaluation metrics: *Efficacy* (Eff.), *Generality* (Gen.), *Locality* (Loc.), *Fluency* (Flu.). For *Locality*, we report the

Table 3: The overall results for single-editing (# Editing=1) and batch-editing (# Editing>1) using HuatuoGPT-o1-8B on MedCF++ and MedVersa datasets.

| # Editing | Method | MedCF++ | | | | | | | | MedVersa | | | | |
|---|---|---|---|---|---|---|---|---|---|---|---|---|---|---|
| | | Eff. | Gen. | Loc. | | | | Flu. | Avg. | Eff. | Gen. | Loc. | Flu. | Avg. |
| | | | | TD | EM | SS | TS | | | | | | | |
| 1 | MEND | 17.74 | 18.23 | 73.58 | 73.09 | 71.28 | 71.55 | 629.71 | 45.18 | 22.77 | 24.40 | 86.90 | **634.22** | 55.24 |
| | MEMIT | 52.92 | 46.47 | _95.94_ | 77.45 | _94.42_ | _92.54_ | 628.60 | 69.89 | 62.33 | 44.43 | 98.77 | 632.54 | 76.08 |
| | MedLaSA | 61.32 | 60.98 | 70.39 | 75.30 | 69.65 | 69.95 | 626.28 | 66.24 | _66.93_ | **65.27** | 86.43 | 625.09 | _76.27_ |
| | RECIPE | _72.98_ | _77.31_ | 91.84 | **99.64** | 92.91 | 91.70 | _652.46_ | _84.58_ | 51.52 | 51.54 | 98.99 | 633.00 | 75.26 |
| | **Ours** | **77.05** | **78.66** | **99.60** | _97.90_ | **98.95** | **98.24** | **653.11** | **88.26** | **69.47** | _63.43_ | **100.00** | _634.05_ | **83.22** |
| 10 | MEND | 15.13 | 16.18 | 54.80 | 57.32 | 53.05 | 51.63 | 630.62 | 34.93 | 20.96 | 22.05 | 68.76 | 636.48 | 45.13 |
| | MEMIT | 52.09 | 46.39 | _89.10_ | 76.47 | 85.42 | 83.67 | 627.76 | 66.45 | _63.97_ | 45.25 | 93.16 | 631.95 | 73.89 |
| | RECIPE | _72.82_ | _76.86_ | 88.80 | _96.24_ | 90.89 | 88.02 | **659.48** | _82.91_ | 51.52 | _51.57_ | 96.34 | **662.74** | _73.94_ |
| | **Ours** | **77.07** | **78.67** | **99.31** | **97.74** | **98.78** | **98.07** | _659.16_ | **88.17** | **69.47** | **63.48** | **99.88** | _662.68_ | **83.17** |
| 50 | MEND | 10.72 | 11.47 | 31.29 | 34.16 | 29.89 | 31.00 | 603.95 | 21.34 | 16.19 | 16.72 | 16.72 | 636.52 | 31.62 |
| | MEMIT | 49.47 | 44.06 | _82.71_ | 74.03 | 77.14 | 77.88 | 628.15 | 62.35 | _65.66_ | 47.47 | 81.81 | 631.25 | _69.19_ |
| | RECIPE | _72.20_ | _75.90_ | 79.31 | _87.38_ | 81.36 | _79.57_ | _659.15_ | 77.98 | 51.52 | _51.46_ | 86.56 | **662.54** | 69.02 |
| | **Ours** | **77.10** | **78.63** | **98.11** | **97.45** | **97.60** | **97.18** | **659.39** | **87.72** | **69.47** | **63.40** | **99.62** | _662.87_ | **83.03** |
| 100 | MEND | 7.45 | 7.54 | 19.69 | 21.91 | 19.49 | 17.87 | 555.39 | 13.62 | 13.62 | 13.90 | 32.66 | 554.89 | 23.21 |
| | MEMIT | 51.40 | 45.77 | _75.82_ | 72.04 | _73.90_ | _73.71_ | 627.77 | 61.23 | _55.71_ | 45.17 | 79.54 | 631.35 | 64.99 |
| | RECIPE | _70.73_ | _74.23_ | 72.25 | _78.62_ | 73.60 | 72.65 | **659.67** | _73.38_ | 51.52 | _51.35_ | 79.55 | **663.31** | _65.49_ |
| | **Ours** | **76.42** | **78.15** | **96.59** | **96.34** | **96.57** | **96.00** | _659.51_ | **86.83** | **69.47** | **63.28** | **99.06** | _663.02_ | **82.72** |

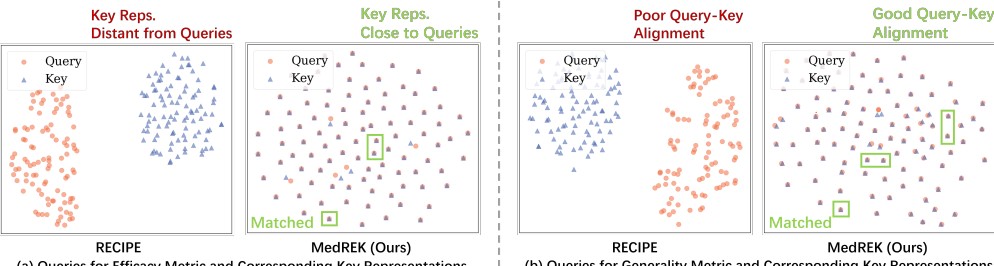

Figure 4: Distribution of query and corresponding key representations (i.e., the keys of the $k$–$v$ pairs in the knowledge base) under a batch of 100 edits using Meditron-7B on the MedCF++ dataset.

(a) Queries for Efficacy Metric and Corresponding Key Representations

(b) Queries for Generality Metric and Corresponding Key Representations

original four sub-metrics on MedCF++, and a simplified overall score on MedVersa. Definitions of each Loc. sub-metric are provided in Appendix A. The weighted average (Avg.) is computed as in MedLaSA (Xu et al., 2024) to capture the trade-off between editing success (Eff. and Gen.) and Loc.

$$\text{Average} = \left( \frac{\text{Eff.}+\text{Gen.}}{2} + \frac{\sum\limits_{m \in \text{Loc.}} m}{|\text{Loc.}|} \right) \Big/ 2. \tag{13}$$

## 5.2 Results on MedCF++ and Medversa (RQ1)

**Single-editing.** From Table 2 and Table 3, we observe that MedREK achieves competitive single-editing performance across both MedCF++ and MedVersa, using different LLM backbones. It outperforms baselines on most sub-metrics, with a notably improved overall average. Interestingly, MedLaSA (Xu et al., 2024) exhibits a significant drop in Locality on both datasets, supporting our earlier claim (Section 3.2) that parameter-modifying methods introduce side effects on unrelated knowledge. In contrast, MedREK performs consistently well across all evaluation setups, highlighting its robustness.

**Batch-editing.** Detailed evaluations with 10, 50, and 100 edits are shown in Table 2 and Table 3. We observe that RECIPE (Chen et al., 2024b) suffers a clear performance drop compared to single-editing, with degradation worsening as the number of edits increases. This suggests that overlapping key representations limit its effectiveness under batch-editing scenarios. In contrast, MedREK consistently performs well across all sub-metrics, achieving the best or second-best results, which highlights the strength of our improved representation model and prompt encoder in handling large-scale edits.

**Remark.** The experimental results on the original MedCF dataset are provided in Appendix D.

## 5.3 Abation Study (RQ2)

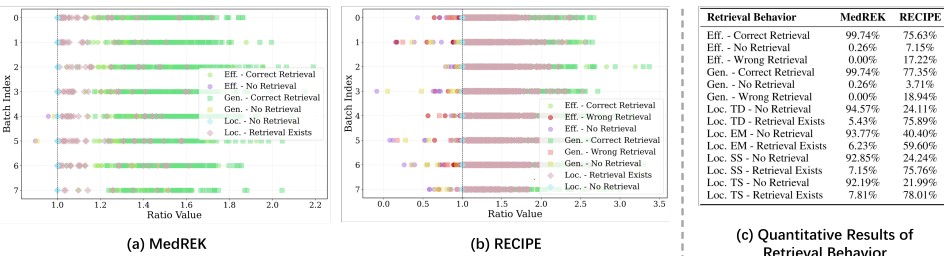

Figure 5: (a-b) Distribution of relative similarity between the query, the ground-truth knowledge, the most similar knowledge and the prototype for test samples in 100-edit batch-editing using Meditron-7B on MedCF++. For Eff. and Gen., x values < 1 indicate incorrect or no retrieval; values > 1 indicate correct retrieval. For Loc., x values = 1 indicate no retrieval (locality preserved); values > 1 imply unintended retrieval. (c) Retrieval statistics across Eff., Gen., and Loc.

To understand the contribution of each proposed component, i.e., the shared query-key MLP and the attention-based prompt encoder, we conduct an ablation study on batch-editing with 100 edits using MedCF++. From Table 4, we observe both components contribute significantly to the final performance of our MedREK. The shared query-key MLP is critical for precise alignment between queries and stored knowledge, enabling more effective retrieval. Without it, performance drops notably due to mismatched representations, highlighting the importance of this component for accurate query-key interaction. The attention mechanism in the prompt encoder also plays a key role by generating high-quality prompts, which further enhances overall editing performance.

Table 4: Ablation of Important Modules on MedCF++ using Meditron-7B.

| # Editing | Variant | MedCF++ | | | | | | | |
|---|---|---|---|---|---|---|---|---|---|
| | | Eff. | Gen. | Loc. | | | | Flu. | Avg. |
| | | | | TD | EM | SS | TS | | |
| 100 | w/o shared MLP | 56.74 | 59.28 | 81.59 | 83.90 | 80.00 | 81.79 | 591.65 | 69.91 |
| | w/o Attn. Prompt Enc. | 73.17 | 75.63 | 97.07 | 94.28 | 96.73 | 96.21 | 590.21 | 85.24 |
| | w/ both (Ours) | 77.96 | 79.88 | 97.76 | 98.20 | 97.57 | 97.19 | 588.01 | 88.30 |

## 5.4 KNOWLEDGE RETRIEVAL EFFECTIVENESS (RQ3)

**Obs1: MedREK achieves better retrieval via precise query-key alignment.** We analyze the distribution of query representations and key representations from the $k$–$v$ pairs in one 100-edit batch. As shown in Figure 4, MedREK aligns query and key representations well for both Efficacy and Generality inputs, indicating precise retrieval. In contrast, RECIPE (Chen et al., 2024b) shows more scattered and distant representations, leading to lower scores in Eff. (68.30 vs. 77.96) and Gen. (70.74 vs. 79.88). These observations suggest that RECIPE struggles to retrieve the correct knowledge, while MedREK benefits from more accurate query-key alignment, resulting in better editing success.

**Obs2: MedREK achieves more accurate and controlled retrieval.** To evaluate retrieval effectiveness, we visualize the distribution of test samples across Eff., Gen., and Loc. metrics in the 100-edit batch setting on MedCF++, along with quantitative retrieval statistics (Figure 5). For Eff. and Gen., the x-axis denotes the ratio between the query's similarity to ground-truth knowledge vs. to the prototype. Values < 1 indicate incorrect or no retrieval. Samples are categorized as correct (green), incorrect, or no retrieval. MedREK achieves a significantly higher rate of correct retrievals than RECIPE (Chen et al., 2024b). For Loc., the x-axis represents the ratio between the query's similarity to the most similar knowledge entry (including prototype) and the prototype itself. Values > 1 indicate unintended retrieval of real knowledge, which harms locality. MedREK shows fewer such cases, indicating better locality preservation. Quantitative results further confirm these observations: MedREK retrieves correct knowledge in nearly all Eff. and Gen. cases while avoiding unwanted retrievals in Loc., in stark contrast to RECIPE. More details are provided in Appendix A.

## 6 CONCLUSION

To address the practical challenge of updating clinical knowledge in medical LLMs, we introduce MedVersa, a benchmark for batch-wise model editing with broad coverage of medical domains. We further propose MedREK, a retrieval-based editing framework tailored for medical LLMs. By incorporating a shared query–key MLP and an attention-based prompt encoder, MedREK enables precise retrieval and effective knowledge updates. Experiments showcase the superiority of MedREK.

ETHICS STATEMENT

This research follows the ICLR Code of Ethics. Our work focuses on developing method for knowledge editing in medical LLMs. The primary goal is to improve the adaptability and reliability of LLMs when incorporating updated or corrected medical information, thereby supporting more accurate and trustworthy downstream applications.

REPRODUCIBILITY STATEMENT

To support reproducibility, we provide a complete description of our proposed MedREK in Section 4.2, with training and evaluation setups detailed in Section 5.1. Hyper-parameter settings are included in Appendix A. Dataset details are described in Section 4.1 and Appendix C.

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

APPENDIX

# A   MORE IMPLEMENTATION DETAILS.

Table 5: Hyper-parameters for MedREK on MedCF++ and MedVersa.

| Hyper-parameter | MedCF++ | MedVersa |
|---|---|---|
| Meditron-7B | | |
| Learning Rate | $1 \times 10^{-5}$ | $1 \times 10^{-5}$ |
| Batch Size | 8 | 8 |
| Repsresentation Dimension | 4096 | 4096 |
| Model Hidden Size | 4096 | 4096 |
| # Prompt Tokens | 3 | 8 |
| # Knowledge Prototype Tokens | 10 | 10 |
| HuatuoGPT-o1-8B | | |
| Learning Rate | $1 \times 10^{-5}$ | $1 \times 10^{-5}$ |
| Batch Size | 8 | 8 |
| Repsresentation Dimension | 4096 | 4096 |
| Model Hidden Size | 4096 | 4096 |
| # Prompt Tokens | 3 | 8 |
| # Knowledge Prototype Tokens | 10 | 10 |

**Training Setup.** We train our MedREK and RECIPE (Chen et al., 2024b) on the training set of MedCF++ and MedVersa. As we observe the checkpoints after around 150 epochs all exhibit a trend of increasing loss for both methods, we stop our training at epoch 200. The hyper-parameters for training and evaluation are kept the same as the baseline methods, except for the number of continuous prompt tokens, which we tune for RECIPE (Chen et al., 2024b). For RECIPE (Chen et al., 2024b) and MedREK, we both use 3 prompt tokens on MedCF++ and 8 prompt tokens on MedVersa. The complete hyper-parameter settings of training for MedREK are shown in Table 5. All experiments are conducted on NVIDIA RTX 5090 GPUs.

**Locality Metrics of MedCF and MedCF++.** Following MedLaSA (Xu et al., 2024), the sub-metrics of Locality in MedCF (Xu et al., 2024) and MedCF++ are defined as follows:

- Target Distribution (TD): Does the editing operation alter the probability distribution of the ground truth tokens?
- Entity Mapping (EM): Does the editing operation solely learn the mapping relationship between head and tail entities?
- Structural Similarity (SS): Does the editing operation influence unrelated knowledge with similar graph structures?
- Textual Similarity (TS): Does the editing operation have an impact on unrelated knowledge that contains similar semantic text?

**Details for Figure 5.** We present the visualization of the distribution of test samples for each metric focusing on the effectiveness of the knowledge retrieval, i.e., whether the correct piece of knowledge is retrieved for Eff. and Gen, and whether the knowledge is "ignored" for Loc. The result is obtained with batch-editing of 100 edits on MedCF++. The y-axis corresponds to the batch index the test samples belonging to.

Recall that the prompt selection criteria for each metric is as follows: For Eff. and Gen., the correct prompt is supposed to be selected. We calculate the similarity between the query and the top-1 similar key of the knowledge entry in the knowledge base, $sim(q, \text{top}_1)$, and the similarity between the query and the knowledge prototype, $sim(q, proto)$. If $sim(q, \text{top}_1) > sim(q, proto)$, we select the prompt that generates $sim(q, \text{top}_1)$. Note that there are three cases here: (1) Correct selection, where the prompt corresponding to $sim(q, \text{top}_1)$ is the target knowledge. (2) Wrong selection, where the prompt corresponding to $sim(q, \text{top}_1)$ is a piece of unrelated knowledge. (3) No selection, where $sim(q, \text{top}_1) = sim(q, proto)$, meaning the most similar entry is the prototype, and no retrieval is

triggered. For Loc., no prompt is supposed to be selected. There are two cases: (1) If $sim(q, \text{top}_1)$ $= sim(q, proto)$, which is the desired behavior, no prompt will be retrieved. (2) If $sim(q, \text{top}_1) >$ $sim(q, proto)$, the query will falsely retrieve a prompt.

We calculate the relative magnitude between $sim(q, \text{top}_1)$, $sim(q, gt)$ (for Eff. and Gen.) and $sim(q, proto)$ to obtain the x value. For editing success (Eff. and Gen.), we present cases of correct retrieval, wrong retrieval, and no retrieval and label the three cases out. For Loc., we do not label correct or wrong, instead, if x = 1, it means the query selects no prompt as desired, and if x > 1, it means the query falsely selects a prompt, which is not desired. For editing success, more samples labeled as Correct Retrieval (green) means better retrieval performance. For Loc., less samples with x > 1 means better retrieval.

**Analysis of MedLaSA.** We find that MedLaSA (Xu et al., 2024) has the following critical design flaw and major weaknesses. First, it violates the spirit of knowledge editing since it considers the rephrase and locality queries for evaluation as known knowledge and applies causal tracing on them to calculate the corresponding layer-wise scaling factors. When testing, it retrieves the scaling factors calculated beforehand with the query text as key. In real-world scenarios, input queries are often unknown or unavailable in advance, making it impractical to apply causal tracing to determine the scaling factors for the adapters. As a result, effective knowledge editing cannot be achieved in practice. Second, despite the utilization of scaling factors to control the impact of adapters on each layer, it does not guarantee precise control. The LoRA (Hu et al., 2022) process may still introduce unintended intervention in model where no modification is desired, thereby affecting unrelated knowledge. In contrast, our method transforms the queries into representations and matches them with the knowledge key representations for prompt retrieval without operations on the queries in advance, making it practical in real-world scenarios. Additionally, we achieve better locality by leaving the original model parameters frozen and only retrieving the prompt when necessary, which is crucial for editing medical LLMs.

## B CONSTRUCTION OF MEDCF++ DATASET.

As mentioned in 4.1, the only existing medical factual knowledge editing benchmark, MedCF (Xu et al., 2024), suffers from design issues that prevent it from supporting batch-editing, which we address by proposing an improved version, MedCF++. Specifically, we remove any records in which the same prompt is used for both Eff. and Loc., as well as any records where Eff. and Loc. share the same prompt across different data entries. As a result, 181 records are removed from the training set, 47 from the validation set, and 47 from the test set. The cleaned dataset MedCF++ avoids prompt overlap to ensure reliable evaluation for batch-editing.

## C CONSTRUCTION OF MEDVERSA DATASET.

Although MedCF++ supports batch-editing, it is still limited in its size and range of medical domains, as most of the entities it selects from the source dataset DRKG are drugs and compounds, causing the majority of the knowledge to fall under the pharmacology domain. To address these limitations, we construct the MedVersa dataset derived from MedMCQA (Pal et al., 2022), which allows for broader coverage of medical knowledge and supports batch-editing.

**Efficacy and Generality Data Construction.** The Efficacy QA pair $(q_e, o_e)$ is used to measure the effectiveness of model editing. We utilize the MedMCQA (Pal et al., 2022) dataset, which contains medical knowledge in the form of multiple-choice questions, each comprising a correct answer and three incorrect options. Since the original "question" field in MedMCQA (Pal et al., 2022) is often expressed as a phrase instead of an interrogative, we first employ Gemini to rewrite it into a clear question form to ensure linguistic clarity and consistency. To construct the Efficacy QA pair, the rewritten question is paired with the correct answer as the ground truth, while one incorrect option is sampled as the counterfactual edit target. For example, as shown in Table 1, the original question in MedMCQA (Pal et al., 2022) "Treatment of multiple carboxylase deficiency" is reformulated into the well-formed interrogative "What is the treatment for multiple carboxylase deficiency?". The original correct answer "Biotin" is retained as the ground truth, while one incorrect option "Thiamine" is selected as the edit target. For the Generality QA pair $(q_g, o_e)$, which evaluates editing effectiveness

Table 6: The prompts for querying Gemini 2.0 Flash.

| Prompts for Querying LLMs |
|---|
| **Question Transformation**: 
 You are given a phrase and its corresponding answer. Please rewrite the phrase into a clear question. 
 Input: 
 Phrase: {original question} 
 Answer: {correct answer} 
 Output only the rewritten question. |
| **Question Rephrase**: 
 Please rephrase the following question using precise medical terminology, ensuring that the original 
 meaning is fully preserved. 
 Question: {question} 
 Output only the rephrased question. |

Table 7: The distribution of medical subjects in MedVersa (%).

| Medical Subject | Percentage (%) |
|---|---|
| Anatomy | 11.22 |
| Microbiology | 10.18 |
| Physiology | 10.08 |
| Surgery | 10.02 |
| Social & Preventive Medicine | 9.97 |
| Gynaecology & Obstetrics | 8.14 |
| Ophthalmology | 7.94 |
| Forensic Medicine | 6.78 |
| Pediatrics | 6.61 |
| ENT | 6.02 |
| Medicine | 2.71 |
| Pathology | 2.62 |
| Pharmacology | 2.05 |
| Biochemistry | 1.98 |
| Orthopaedics | 0.94 |
| Radiology | 0.82 |
| Psychiatry | 0.64 |
| Anaesthesia | 0.47 |
| Dental | 0.44 |
| Skin | 0.37 |

on similar questions, we employ Gemini to rephrase the Efficacy question. The prompts for querying Gemini are shown in Table 6.

**Locality Data Construction.** Locality aims to evaluate whether the edited model preserves unrelated knowledge, and the model is expected to generate the same answer as before editing. To construct the Locality data, we sample a different entry from MedMCQA (Pal et al., 2022) than the one used for Efficacy, but within the same medical subject, leveraging the "subject name" field in MedMCQA (Pal et al., 2022). The original question is then converted into a standard interrogative, with the correct answer as the ground truth for the Locality question. For example, as shown in Table 1, the Locality question "At what age does purposeful movement typically start in infants?" represents a different piece of knowledge from the Efficacy question, while sharing the same medical subject "Pediatrics". This design assesses the reliability of the model in preserving unrelated but same-domain knowledge after editing.

**Comparison of Medical Subjects in MedVersa and MedCF++.** Table 7 and Table 8 show the full distribution of medical subjects in MedVersa and MedCF++. Compared with MedCF++, which is dominated by Pharmacology, MedVersa exhibits a more balanced coverage across subjects. In addition, MedVersa includes 8 subjects that are absent in MedCF++. These underscore the broader coverage of medical domains of MedVersa, enabling a more comprehensive evaluation of medical knowledge editing.

Table 8: The distribution of medical subjects in MedCF++ (%).

| Medical Subject | Percentage (%) |
|---|---|
| Pharmacology | 71.78 |
| Biochemistry | 6.96 |
| Pathology | 5.24 |
| Medicine | 4.48 |
| Anatomy | 2.98 |
| Gynaecology & Obstetrics | 1.94 |
| Physiology | 1.86 |
| Psychiatry | 1.44 |
| Ophthalmology | 1.31 |
| Skin | 0.73 |
| Orthopaedics | 0.73 |
| Surgery | 0.55 |

Table 9: The overall results for single editing using Meditron-7B on original MedCF dataset.

| Editing Type | Method | MedCF | | | | | | | |
|---|---|---|---|---|---|---|---|---|---|
| | | Eff. | Gen. | Loc. | | | | Flu. | Avg. |
| | | | | TD | EM | SS | TS | | |
| Single Editing | FT | 65.97 | 65.36 | 48.91 | 50.39 | 48.13 | 46.25 | 327.76 | 57.04 |
| | LoRA | 72.19 | 71.80 | 92.29 | 91.11 | 91.36 | 92.42 | 572.33 | 81.90 |
| | MEND | 22.87 | 22.93 | 71.16 | 71.21 | 71.03 | 72.29 | 428.38 | 47.16 |
| | ROME | 72.69 | 72.91 | 92.79 | 61.80 | 90.06 | 86.93 | 559.82 | 77.84 |
| | MEMIT | **83.10** | **83.23** | 95.01 | 62.62 | 92.99 | 90.50 | 563.31 | 84.22 |
| | MedLaSA | 72.37 | 71.06 | 95.71 | 94.84 | 95.04 | 94.90 | 582.80 | 83.42 |
| | RECIPE | 72.34 | 75.40 | 93.63 | **97.09** | 92.91 | 93.91 | 573.97 | 84.13 |
| | Ours | 77.91 | 79.83 | **99.45** | 96.80 | **99.36** | **98.75** | **586.34** | **88.73** |

# D  MORE EXPERIMENTAL RESULTS.

**Results of Single Editing on MedCF.** In Table 9,we present the results of single-editing using Meditron-7B on the original MedCF (Xu et al., 2024) dataset, which contains duplicate prompts for Efficacy and Locality in a single record. MedREK performs consistently well on it, and obtains better result especially in Loc.-EM on the cleaned dataset MedCF++ as shown in Table 2. This supports our earlier claim (Section 4.1) that duplicated prompts in MedCF (Xu et al., 2024) can lead to unreliable evaluations.

**Edit Time of Different Methods.** To compare the efficiency of different methods, we conduct single-edit experiments using Meditron-7B (Li et al., 2023) on MedVersa and report the average edit time in Table 10. We observe that MedREK and RECIPE (Chen et al., 2024b)—both retrieval-based methods—significantly outperform parameter-modifying methods. In particular, MedLaSA (Xu et al., 2024) is a parameter-modifying method that leverages LoRA (Hu et al., 2022) finetuning to perform edits. However, it requires 70 epochs of finetuning for each single edit, which results in substantially longer editing time. This makes it less efficient and impractical for scenarios requiring rapid or frequent knowledge updates.

# E  TRAINING LOSS OF MEDREK

Following RECIPE (Chen et al., 2024b), the loss functions are defined as follows:

$$\mathcal{L}_{\text{eff}}^{(i)} = -\log \hat{f}\_\theta \left( o^{(i)}\_e \mid p^{(i)} \oplus f\_\text{emb}(q^{(i)}\_e) \right) \qquad (1)$$

$$\mathcal{L}_{\text{gen}}^{(i)} = -\log \hat{f}\_\theta \left( o^{(i)}\_g \mid p^{(i)} \oplus f\_\text{emb}(q^{(i)}\_g) \right) \qquad (2)$$

Table 10: Average edit time taken across different methods for single-editing using Meditron-7B on MedVersa.

| Method | Edit Time (s) |
|---|---|
| MEND | 3.301 |
| MEMIT | 18.239 |
| MedLaSA | 16.787 |
| RECIPE | **0.006** |
| MedREK (Ours) | 0.012 |

$$\mathcal{L}_{\text{loc}}^{(i)} = \text{KL}\left(f\_\theta(q^{(i)}\_l) \parallel \hat{f}\_\theta\left(p^{(i)} \oplus f\_\text{emb}(q^{(i)}\_l)\right)\right) \qquad (3)$$

where $f_\theta$ is the large language model (LLM) to be editied, and $\hat{f}_\theta$ is $f_\theta$ with the embedding layer $f_{\text{emb}}$ removed. The editing loss is then defined as $\mathcal{L}_{\text{edit}} = \frac{1}{B}\sum_{i=1}^{B}\left(\mathcal{L}_{\text{eff}}^{(i)} + \mathcal{L}_{\text{gen}}^{(i)} + \mathcal{L}_{\text{loc}}^{(i)}\right)$. The contrastive learning loss for prompt learning are defined as follows:

$$\mathcal{L}_{\text{no}}^{(i)} = \delta\left(\mathbf{z}_{q_e}^{(i)}, \mathbf{z}_v^{(i)}, R\right) + \delta\left(\mathbf{z}_{q_g}^{(i)}, \mathbf{z}_v^{(i)}, R\right) \qquad (4)$$

$$\mathcal{L}_{\text{so}}^{(i)} = \delta\left(\mathbf{z}_{q_l}^{(i)}, \mathbf{z}_{pt}, R\right) + \delta\left(\mathbf{z}_{q_e}^{(i)}, \mathbf{z}_{pt}, R \setminus v_i\right) + \delta\left(\mathbf{z}_{q_g}^{(i)}, \mathbf{z}_{pt}, R \setminus v_i\right) \qquad (5)$$

$$\mathcal{L}_{\text{contra}} = \frac{1}{b}\sum_{i=1}^{b}\left(\mathcal{L}_{\text{no}}^{(i)} + \mathcal{L}_{\text{so}}^{(i)}\right) \qquad (6)$$

where $R = \{\mathbf{z}_v^{(i)}\}_{i=1}^{b} \cup \{r_\Theta\}$ and $R \setminus \mathbf{z}_v^{(i)} = R \setminus \{\mathbf{z}_v^{(i)}\}$. $\mathbf{z}_v^{(i)}$ is the representation of the editing knowledge triple $\text{KN}_e^{(i)}$ transformed through Equation (9) in the paper. The query representations $\mathbf{z}_{q_e}^{(i)}$, $\mathbf{z}_{q_g}^{(i)}$, and $\mathbf{z}_{q_l}^{(i)}$ for $q_e^{(i)}$, $q_g^{(i)}$, and $q_l^{(i)}$ are attained via Equation (8) in the paper, respectively. $\delta$ is the InfoNCE loss [2], formulated as:

$$\delta\left(q, \text{KN}_e^+, \{\text{KN}_e^{(i)}\}_{i=1}^{n}\right) = -\log\left(\frac{\exp(q \cdot \text{KN}_e^+/\tau)}{\sum_{i=1}^{n}\exp(q \cdot \text{KN}_e^{(i)}/\tau)}\right) \qquad (7)$$

where $\tau$ is the temperature, typically set to 1 by default. Finally, the total training loss is defined as $\mathcal{L}_{\text{total}} = \mathcal{L}_{\text{contra}} + \mathcal{L}_{\text{edit}}$.

# F USE OF LLMs

We leverage Gemini 2.0 Flash for constructing the Efficacy, Generality, and Locality data in MedVersa, as detailed in C. For Efficacy and Locality questions, the original phrases from MedMCQA (Pal et al., 2022) are transformed into interrogatives using Gemini. For Generality questions, we employ Gemini to rephrase the corresponding Efficacy questions. The prompts used for querying the LLM are provided in Table 6. We took measures to ensure that the rewritten and rephrased questions maintained their original meaning. We used ChatGPT solely to improve the fluency and clarity of writing. All content was written by the authors, and the model served only as a language assistant.

