# OpenReview forum: "MedREK: Retrieval-Based Editing for Medical LLMs with Key-Aware Prompts"
_ICLR.cc/2026/Conference — ICLR 2026 Conference Withdrawn Submission_

### Official Review · Reviewer_7Vo1 · 2025-10-31

**Soundness:** 2
**Presentation:** 2
**Contribution:** 2
**Rating:** 4
**Confidence:** 2

**Summary:**

This paper studies the problem of knowledge editing in the medical domain and introduces a retrieval-based framework designed for batch editing, along with a new benchmark for evaluating batch edit performance. To mitigate the issue of overlapping keys, the proposed method embeds subject–relation pairs and employs an attention-based encoder to generate soft prompts. Experimental results demonstrate that the proposed approach achieves state-of-the-art performance across multiple evaluation metrics.

**Strengths:**

1. The motivation of batch edits is reasonable and the proposed benchmark can potentially inspire future studies.
2. The experiment is comprehensive and demonstrates the proposed model’s effectiveness.

**Weaknesses:**

1. Regarding the methodology, it appears that this paper largely follows the design of RECIPE, which also formulates knowledge editing as training retrieved soft prompts. Although the authors introduce some specific modifications, such as using a pretrained RoBERTa encoder and a shared MLP projection, these seem to be more engineering adjustments rather than fundamentally new technical contributions.
2. The idea of the attention-based prompt encoder is also unclear to me. Since the attention is performed over a single embedding, the resulting soft prompt essentially comes from distributing attention scores over the projected value embedding. I am uncertain about the underlying criterion here and wonder how the performance would compare if the model simply projected z_v into T embeddings directly, without the attention operation.
3. Moreover, it remains unclear how these proposed techniques specifically address the challenges of batch editing.
4. As for the ablation study, `w/o shared MLP` means removing the MLP (only use H?) or using separate MLPs? What does `w/o Attn Prompt Enc.` represent? How do you obtain the soft prompts in this variant?

**Questions:**

See weaknesses.

---

> ### Author Response · Authors · 2025-11-26
> **To Reviewer 7Vo1**
>
> Thanks for the constructive comments on our paper. Please kindly find our response to your comments below.
>
> Q1. Comparison with RECIPE.
> > While it is true that our method adopts the retrieval-based prompt tuning framework similar to RECIPE, the motivation and technical design of MedREK are driven by the unique challenges of the medical domain.
> >
> > As shown in Section 3.3, RECIPE suffers from representation overlap in the retrieval space and often fails in batch editing scenarios, particularly in terms of Locality, which is critical for medical applications (as indicated by the results in Table 2).
> >
> > To address these issues, MedREK introduces the simple yet effective shared query-key MLP to unify the representation space for more precise retrieval, and an attention-based prompt encoder to generate informative prompts that guide accurate edits. These components are specifically designed to preserve locality and improve performance in realistic medical batch-editing scenarios.
>
> Q2. Attention-based prompt encoder.
>
> > In our attention-based prompt encoder, each batch sample has a single key–value embedding derived from the knowledge representation. While this means there is no sequence-level attention over multiple keys, the multi-query design still allows each prompt token to extract distinct aspects of the same knowledge vector. Specifically:
> > - Each prompt token has its own query $q_t = W_q^{(t)} z_v, \quad t = 1, \dots, T$.
> > - The attention computation $p_t = softmax(q_tk^{T})v$ acts as a learned adaptive gating, modulating how much each token extracts from the shared key/value vector.
> > - This enables per-token specialization: different prompt tokens can emphasize different directions or features of the knowledge representation, even though the underlying key/value is the same.
>
>
> Q3. How proposed techniques specifically address the challenges of batch editing.
> > Batch editing in MedREK is enabled by the retrieval-based framework, which maintains a knowledge base allowing multiple knowledge entries to be edited simultaneously. In contrast, the prior medical knowledge editing method MedLaSA relies on modifying model parameters, which inherently limits it to single edits and prevents it from handling batch editing scenarios.
>
> Q4. Ablation study setup.
> > `w/o shared MLP` means using seperate MLPs for queries and keys.
> >
> >
> > In the ablation variant `w/o Attn Prompt Enc.`，we remove the attention mechanism from the prompt encoder and replace it with a feed-forward module that generates each prompt token independently.
> >
> > Specifically, instead of using Q–K–V and multi-head attention, this variant employs $T$ independent MLPs, one for each prompt token. Given the knowledge representation  $z_{v}$, each soft prompt token is computed as:
> $$p\_{t} = \mathrm{MLP}\_{t}(z\_{v}),\quad t=1,\ldots,T.$$
> > These MLPs do not share parameters and therefore preserve the per-token parameterization used in the attention-based encoder, while eliminating any token–token interaction introduced by attention. The resulting soft prompts are obtained by stacking the $T$ independently generated tokens.

---

### Official Review · Reviewer_waxX · 2025-11-01

**Soundness:** 3
**Presentation:** 2
**Contribution:** 3
**Rating:** 4
**Confidence:** 5

**Summary:**

**Problem & Motivation**

The paper addresses the unreliability of LLMs in high-stakes medical applications. Due to the rapid evolution of medical science and potential training data errors, LLMs often generate outdated or inaccurate information (hallucinations). While "model editing" (updating model knowledge without full retraining) is a potential solution, standard methods face critical limitations in the medical domain.
The authors identify three categories of model editing, but focus on the critical flaws of the two parameter-modifying approaches:
1. *Locate-then-edit*: These methods directly modify a small subset of the model's weights. The paper argues this is ill-suited for medicine because it "compromises locality," meaning an edit to one fact can unintentionally alter unrelated knowledge.
2. *Meta-learning-based*: These methods manipulate gradients to perform global parameter updates.
3. *Retrieval-based editing*: This approach avoids altering the original LLM parameters by storing new knowledge in an external memory (a key-value store) and retrieving it at inference time; e.g., RECIPE uses continuous prompt learning to prefix knowledge to the input query. This is the paper's preferred approach as it inherently preserves locality.

The paper's method is motivated by two specific failures of existing retrieval-based editors in the medical domain:
- Medical knowledge contains many textually similar but factually distinct items (e.g., "drug-drug interaction with Clofedanol" vs. "drug-drug interaction with Camostat"). This causes representation overlap in the retrieval space, leading the model to retrieve the wrong fact and reducing editing accuracy.
- Lack of batch-editing evaluation: existing benchmarks (like MedCF) focus on single-fact edits. However, real-world medical updates (e.g., new guidelines, retracted studies) often require updating multiple related facts at once (batch-editing).

**Method**

- **MedVersa.** To address the lack of a suitable benchmark, the authors first create MedVersa. It is constructed from the MedMCQA dataset and is designed to be superior to the previous MedCF benchmark.
   - Batch-editing support. It fixes a flaw in MedCF where "efficacy" (what you edit) and "locality" (what should not change) prompts overlapped, which made true batch-editing evaluation not possible.
   - Broader coverage. It spans 20 medical subjects, instead of 12 of MedCF.

- **MedREK (Medical Retrieval-based Editing with Key-aware prompts)**, the main contribution, a new retrieval-based editing framework. Its mechanism is trained in two phases. First, a knowledge base is built where each new fact (s, r, o) is encoded into a key-value pair: a Shared Query-Key MLP (MLP_qk) encodes the (s, r) pair to create the Key, and an Attention-based Prompt Encoder encodes the full (s, r, o) triplet to create the Value. A user's query is encoded by the same MLP_qk. If the query vector matches a Key (and is not an unrelated query, as determined by a threshold), the corresponding Value embedding is retrieved and prepended to the query, thus acting as a continuous prompt. The total loss L_total = L_edit + L_contra trains the modules for these two distinct goals:
- L_edit: An editing loss that trains the Prompt Encoder to generate "Values" (continuous prompts) that successfully guide the LLM to the new answer.
- L_contra: A contrastive loss (InfoNCE) that trains the Shared MLP_qk to pull matching query/key pairs together and push mismatched pairs apart, mitigating the representation overlap problem.

**Experimental setup**

- The framework is evaluated on two medical-domain LLMs: Meditron-7B (LLaMA2-based) and HuatuoGPT-01-8B (LLaMA3-based).
- Baselines. MedREK is compared against:
   - Parameter-based: MEND, MEMIT, and MedLaSA (a medical-specific editor).
   - Retrieval-based: RECIPE (the previous state-of-the-art retrieval editor).
- Benchmarks: MedCF++ and the new MedVersa.

**Key findings**

- Superior performance. MedREK achieves the highest average scores across all metrics (Efficacy, Generality, Locality) on both benchmarks and both LLMs.
- Batch-editing success. MedREK's performance remains high even in batch-editing (10, 50, 100 edits), while the baseline retrieval-editor (RECIPE) sees its performance degrade significantly as the batch size increases.
- Locality. Parameter-based methods (MedLaSA, MEMIT) show a significant drop in Locality, validating the paper's motivation for using a retrieval-based approach.

**Strengths:**

- The paper addresses knowledge editing within the medical domain. By focusing on the "locality" requirement—ensuring edits do not corrupt unrelated knowledge—it tackles a critical safety and reliability problem specific to high-stakes applications.
- Dual contribution. The authors provide both a new resource and a new method.
- MedREK ablations. The paper includes an ablation study on MedREK's two primary components (the shared query-key MLP and the attention-based prompt encoder). This analysis, as referenced in the "Key Findings" and experimental setup, aims to validate the authors' design choices by isolating the contribution of each module to the framework's overall performance.

**Weaknesses:**

- *Limited comparison.* The results position MedREK primarily against RECIPE. While RECIPE is a strong baseline, the claim of SOTA performance would be substantially stronger if MedREK were benchmarked against a wider array of recent retrieval-based editors. The conclusion that MedREK solves the retrieval problem for medical text is contingent on RECIPE being a sufficient proxy for all retrieval-based methods.
- *Absence of statistical significance testing.* The presented results appear robust. However, for a paper making strong claims of superiority, the lack of statistical significance testing (e.g., p-values from t-tests or similar) is a methodological omission. It is unclear if the reported average performance gains are statistically significant across the entire test set or if they are driven by a high-leverage subset of the benchmark.
- *Prompt encoder motivation.* While the prompt encoder uses multi-head attention to generate prompt tokens, it is not fully transparent from the text or ablation exactly why attention-based prompts yield such significant locality gains. More interpretability analysis (e.g., what information is captured by learned prompts) would enhance understanding and trust in the mechanism.
- *Lack of cross-domain generalization results.* The evaluation is tightly focused on medical LLMs and medical datasets. Experiments on out-of-domain tasks could demonstrate the generality or limits of the approach, especially since the core architectural ideas (retriever + prompt generation) are not specific to the medical domain.
- *Limited model generalizability.* The framework is only validated on two medical LLaMA-based LLMs (Meditron-7B and HuatuoGPT-01-8B). This is a very small model pool, and it remains unproven how MedREK would perform with other model architectures or non-medical-domain backbones.
- *Questionable encoder choice.* The shared query-key MLP is built on a general-purpose RoBERTa. Given the paper's strong emphasis on the medical domain, the failure to test or justify this choice against domain-specific encoders is a clear methodological gap. A domain-specific encoder might inherently resolve some of the "representation overlap" issues.
- *Unspecified efficiency.*  The paper does not provide a clear analysis of the computational overhead of MedREK, as well as exact VRAM requirements.
- *Presentation quality.* The paper suffers from presentation issues that hinder readability. The submission retains a template title ("FORMATTING INSTRUCTIONS..."), and the methodological description (particularly the interaction between the two encoders) is not always easy to parse. Different typos have been identified, e.g., missing punctuation in Section 5.1.
- *No stated data/code sharing plan.* The paper lacks a clear commitment or plan for sharing the MedVersa benchmark or the MedREK implementation code, even for peer review. This omission hinders reproducibility and prevents independent verification of the experimental claims.

**Questions:**

- Could the authors provide targeted case studies of failed edits or unintended locality violations, particularly in batch-editing?
- How sensitive is the performance to the selection of RoBERTa as the encoder for keys/queries? Would other encoders or domain-specific pretraining yield different trade-offs?
- What steps were taken to ensure that the automatic rewriting and rephrasing steps (using Gemini) did not introduce subtle semantic distortions? Is there any human (expert) validation of sampled MedVersa questions and counterfactuals?

---

> ### Author Response · Authors · 2025-11-26
> **To Reviewer waxX**
>
> Thanks for the constructive comments on our paper. Please kindly find our response to your comments below.
>
> Q1. Limited comparison.
>
> > To the best of our knowledge, RECIPE is currently the most recent and strongest retrieval-based knowledge-editing method publicly available for LLMs. If the reviewer can suggest additional retrieval-based editors that they consider relevant or more recent, we would be glad to include those methods and conduct further comparisons in the revised version.
>
> Q2. Absence of statistical significance testing.
> > We thank the reviewer for raising the point about statistical significance testing. Existing knowledge-editing benchmarks are based on fixed, deterministic evaluation sets rather than stochastic sampling procedures, so prior work has rarely reported formal hypothesis tests in this setting. That said, we agree that providing a measure of variability can help the reader better interpret differences between methods, especially when performance gaps are small. In our case, many of the improvements are relatively large and consistent across metrics and datasets, which already suggests strong practical significance. Nevertheless, in the revised version, we will additionally report statistical analyses to quantify the robustness of our gains and address the reviewer’s concern more thoroughly.
>
> Q3. Lack of cross-domain generalization results.
> > Our work is intentionally scoped to medical-domain knowledge editing, and we clearly state in the paper that MedREK is designed for medical LLMs and medical factual updates. Cross-domain generalization is therefore beyond the scope of this study.
>
>
> Q4. Case studies of failed edits or unintended locality violations.
> > Please refer to Figure 2(b), where we provide an example of a failed edit caused by textually similar but factually unrelated knowledge entries. This illustrates an instance of unintended locality violation.
>
> Q5. Performance of domain-specific encoders compared to RoBERTa.
> > We have conducted experiments using Meditron-7B on MedCF++ to compare the effect of different encoders, i.e., RoBERTa and BioBERT.
> >
> > | Encoder      | # Editing      | Eff.   | Gen.   | Loc.-TD | Loc.-EM | Loc.-SS  | Loc.-TS | Flu.   | Avg.   |
> > |:-----------:|:----------:|:----------:|:------:|:------:|:------:|:------:|:------:|:------:|:------:|
> > | RoBERTa | 1       | 78.50  | 80.61  | 99.42  | 98.96  | 99.34   | 98.67 | 587.00  |  89.33
> > | BioBERT | 1     | 81.25  | 83.28  | 99.72  | 99.42  | 99.43  | 99.29 | 590.03  |90.86
> > | RoBERTa | 10     | 78.52  | 80.63  | 99.33  | 98.89  | 99.35  | 98.51 | 589.56  |89.30
> > | BioBERT | 10     | 81.28  | 83.30  | 99.72 | 99.36  | 99.43  | 99.19 | 587.49  |90.86
> > | RoBERTa | 50     | 78.54  | 80.61 | 98.55  | 98.70  | 98.80  | 97.87 | 589.79  | 89.03
> > | BioBERT | 50     | 81.28  | 83.30  | 99.72  | 99.36  | 99.43  | 99.19 | 587.49  | 90.54
> > | RoBERTa | 100    | 77.96  | 79.88  | 97.76 | 98.20  | 97.57  | 97.19 | 588.01  |  88.30
> > | BioBERT | 100          | 80.65  | 82.53  | 98.26  | 98.69  | 98.24  | 98.03 | 588.99  | 89.95
> >
> > The result shows that employing medical domain-specific BioBERT can indeed improve Efficacy and  Generality by approximately 3%. It also improves Locality slightly by approximately 1%.
> >
> > We appreciate the suggestion and will integrate the medical-specific encoder to our method.
>
> Q6. Unspecified efficiency.
> > We have provided the comparison of edit efficiency across different methods in Appendix D. We will include the VRAM requirements in the revised version.
>
> Q7. Presentation quality.
> > We will improve the clarity of the methodological description, particularly regarding the interaction between the two encoders.
> >
> >Regarding the typos mentioned (e.g., in Section 5.1), we carefully reviewed the current version but were unable to locate them. Could the reviewer kindly specify the exact line numbers where these typos occur? This will help us correct them in the revised manuscript.
>
> Q8. Validation of constructed data.
>
> > We conducted manual verification of all rewritten questions generated by Gemini. Due to the lack of access to licensed medical professionals, the verification was performed by the authors, who checked the generated interrogative questions for medical terminology correctness and consistency with the original MedMCQA entries. We will explicitly describe this verification procedure in the revised paper.

---

### Official Review · Reviewer_n5UC · 2025-11-02

**Soundness:** 2
**Presentation:** 3
**Contribution:** 1
**Rating:** 2
**Confidence:** 4

**Summary:**

This paper addresses the limitations of medical Large Language Models (LLMs) in generating outdated or inaccurate information due to rapid medical knowledge evolution and training data errors. It proposes two core contributions: (1) MedVersa, a medical knowledge editing benchmark that supports both single and batch-editing scenarios, with broader coverage of 20 medical subjects (vs. 12 in prior MedCF++) to avoid over-reliance on pharmacology; (2) MedREK, a retrieval-based editing framework integrating a shared query-key MLP (for precise query-knowledge alignment) and an attention-based prompt encoder (for knowledge-specific prompts). Experiments on MedCF++ and MedVersa with Meditron-7B and HuatuoGPT-o1-8B show MedREK outperforms baselines (e.g., MEND, MEMIT, RECIPE) across Efficacy, Generality, and Locality metrics, providing the first validated batch-editing solution for medical LLMs.

**Strengths:**

1. It pioneers the construction of a batch-editing benchmark (MedVersa) for medical LLMs, addressing the gap in prior benchmarks (e.g., MedCF, MedCF++) that only support single-editing. MedVersa’s 20 medical subjects (including Pediatrics and Social & Preventive Medicine) fill the void of narrow domain coverage in existing benchmarks.
2. It innovatively adapts retrieval-based editing to the medical domain, designing a shared query-key MLP to resolve representation overlap (a critical issue in medical knowledge retrieval) and an attention-based prompt encoder to generate fine-grained prompts, overcoming the limitations of parameter-based editing (e.g., MedLaSA) that harms Locality.
3. Experimental design is rigorous: it validates MedREK on two medical LLMs (Meditron-7B, HuatuoGPT-o1-8B) and two benchmarks (MedCF++, MedVersa), covering single (1-edit) and batch (10/50/100-edit) scenarios. Ablation studies confirm the individual contributions of the shared query-key MLP and attention prompt encoder.
4. Benchmark construction is systematic: it first refines MedCF into MedCF++ to eliminate prompt overlap between Efficacy and Locality, then expands to MedVersa using MedMCQA, ensuring each data entry includes Efficacy/Generality/Locality QA pairs for comprehensive evaluation.

**Weaknesses:**

1. The paper uses Gemini 2.0 Flash to rewrite MedMCQA’s original phrases into interrogative questions (for Efficacy/Generality) but provides no evidence of semantic consistency or medical term accuracy. There is no mention of manual verification (e.g., by medical professionals) or automatic metrics (e.g., BERTScore) to rule out ambiguity or term misuse, which may undermine MedVersa’s reliability.
2. RoBERTa is a general pre-trained model, but the paper does not compare it with medical-specific models (e.g., BioBERT, PubMedBERT) in terms of embedding quality for medical text. Without this comparison, it cannot be confirmed that RoBERTa is optimal for capturing medical knowledge representations.
3. Batch-editing is only tested on 10/50/100 edits. The paper does not explore larger batch sizes (e.g., 500 edits) to assess performance degradation, nor does it analyze the scalability of MedREK when the knowledge base expands to millions of medical facts (e.g., whether retrieval speed degrades and if index optimizations like FAISS are needed).
4. The ethics section only mentions compliance with the ICLR Code of Ethics but lacks details: (1) whether MedVersa’s source data (MedMCQA) contains patient-related information and has undergone de-identification; (2) whether ethical committee (IRB) approval was obtained for dataset construction; (3) how user medical queries are protected during MedREK’s deployment.

**Questions:**

see the weaknesses

**Details Of Ethics Concerns:**

1. Unclear data privacy protection: The paper does not specify whether MedVersa’s source dataset (MedMCQA) contains patient-related information (e.g., medical records). If such information exists, there is no statement on whether de-identification (e.g., removing personal identifiers) was performed, which violates medical data privacy protection norms.
2. Lack of ethical approval documentation: It does not mention whether the construction of MedVersa (including data collection and processing) and the use of pre-trained medical LLMs (Meditron-7B, HuatuoGPT-o1-8B) obtained approval from an Institutional Review Board (IRB), which is a core requirement for medical-related research involving human-derived data.
3. Unaddressed deployment privacy risks: MedREK is intended for clinical scenarios, but the paper provides no plan for protecting user medical queries (e.g., symptoms, medical history) during deployment (e.g., end-side retrieval, data encryption), posing potential risks of sensitive information leakage.

---

> ### Author Response · Authors · 2025-11-26
> **To Reviewer n5UC**
>
> Thanks for the constructive comments on our paper. Please kindly find our response to your comments below.
>
> Q1. Concerns regarding the constructed data’s medical term accuracy and the ethical/privacy safeguards of MedVersa.
>
> > We would like to clarify that we did conduct manual verification of all rewritten questions. Due to the lack of access to licensed medical professionals, the verification was performed by the authors, who checked the generated interrogative questions for medical terminology correctness and consistency with the original MedMCQA entries. We will explicitly describe this verification procedure in the revised paper. We can further include an automatic metric (e.g., BERTScore) to quantify consistency.
> >
> > MedVersa is constructed entirely from MedMCQA, which is a publicly available, multiple-choice medical knowledge dataset. Importantly, MedMCQA does not contain any patient-level data, medical records, or personal identifiers. Therefore, no de-identification is required, and the dataset poses no medical privacy risk.
> >
> > We will clarify this in the Ethics section.
>
> Q2. Compare RoBERTa with medical-specific models (e.g., BioBERT, PubMedBERT).
>
> > We have conducted experiments using Meditron-7B on MedCF++ to compare the effect of different encoders, i.e., RoBERTa and BioBERT.
> > | Encoder      | # Editing      | Eff.   | Gen.   | Loc.-TD | Loc.-EM | Loc.-SS  | Loc.-TS | Flu.   | Avg.   |
> > |:-----------:|:----------:|:----------:|:------:|:------:|:------:|:------:|:------:|:------:|:------:|
> > | RoBERTa | 1       | 78.50  | 80.61  | 99.42  | 98.96  | 99.34   | 98.67 | 587.00  |  89.33
> > | BioBERT | 1     | 81.25  | 83.28  | 99.72  | 99.42  | 99.43  | 99.29 | 590.03  |90.86
> > | RoBERTa | 10     | 78.52  | 80.63  | 99.33  | 98.89  | 99.35  | 98.51 | 589.56  |89.30
> > | BioBERT | 10     | 81.28  | 83.30  | 99.72 | 99.36  | 99.43  | 99.19 | 587.49  |90.86
> > | RoBERTa | 50     | 78.54  | 80.61 | 98.55  | 98.70  | 98.80  | 97.87 | 589.79  | 89.03
> > | BioBERT | 50     | 81.28  | 83.30  | 99.72  | 99.36  | 99.43  | 99.19 | 587.49  | 90.54
> > | RoBERTa | 100    | 77.96  | 79.88  | 97.76 | 98.20  | 97.57  | 97.19 | 588.01  |  88.30
> > | BioBERT | 100          | 80.65  | 82.53  | 98.26  | 98.69  | 98.24  | 98.03 | 588.99  | 89.95
> >
> > The result shows that employing medical domain-specific BioBERT can indeed improve Efficacy and  Generality by approximately 3%. It also improves Locality slightly by approximately 1%.
> >
> > We appreciate the suggestion and will integrate the medical-specific encoder to our method.
>
> Q3. Extend to larger batch sizes, e.g. 500 edits.
> > As shown in Figure 3\(c), the test sets of MedVersa and MedCF++ contain only 1000 and 755 samples respectively, which inherently limits the maximum feasible batch size. Therefore, we evaluated batch editing up to 100 edits, which already corresponds to editing 10–13% of the entire test set at once. Moreover, as acknowledged in Strength 3, our experimental design is rigorous. Given this, we are unsure why the lack of a 500-edit experiment is considered a weakness, especially when current medical knowledge-editing benchmarks do not provide larger-scale datasets (e.g., millions of facts). Nevertheless, to further demonstrate scalability, we can add an additional experiment with a 500-edit batch.

---

### Note · Authors · 2026-01-09

I have read and agree with the venue's withdrawal policy on behalf of myself and my co-authors.